# Assessing the impact of the Gamma variant on COVID-19 patient admissions in a southern Brazilian tertiary hospital—A comparison of dual pandemic phases

Natalia R. Domino[1]☯, Bruna A. Lapinscki[2]☯, Felipe Zhen[1], Guilherme Yamaguto[1], Emanueli C. S. Costa[1], Vitor L. Moriya[1], Luciane A. Pereira[3], Ricardo Petterle[4], Meri B. Nogueira[3], Sonia M. Raboni[1,4]*

1 Infectious Diseases Division, Hospital de Clínicas, Universidade Federal do Paraná, Curitiba, Brazil,
2 Microorganisms Research and Molecular Biology Laboratory, Universidade Federal do Paraná, Curitiba,
Brazil, 3 Virology Laboratory, Universidade Federal do Paraná, Curitiba, Brazil, 4 Department of Integrative
Medicine, Universidade Federal do Paraná, Curitiba, Brazil

☯ These authors contributed equally to this work.
* sraboni@ufpr.br

**Data Availability Statement:** All relevant data are within the paper and its Supporting Information files.

## Abstract

Since the first case of COVID-19, Brazil has undergone infection waves with distinct characteristics. The description of new variants has alerted the emergence of more contagious or virulent viruses. The variant of concern Gamma emerged in Brazil and caused an epidemic wave, but its spread outside the country was limited. We report the clinical-epidemiological profile of hospitalized patients with COVID-19 by comparing two periods. A retrospective cohort study was performed. The primary outcome was to assess individuals with COVID-19 admitted in wards and intensive care units at the academic hospital of the Federal University of Parana (CHC-UFPR) between March 2020 and July 2021, correlating demographic, clinical-epidemiologic, and survival data with the most prevalent viral variant found in each period. We used Kaplan-Meier analysis to estimate the probability of survival and ROC curves to evaluate laboratory tests to find a cutoff point for poor outcomes. Data from 2,887 individuals were analyzed, 1,495 and 1,392 from the first and second periods, respectively. Hospitalization predominated among males in both periods, and the median age was significantly lower in the second one. The frequency of comorbidities was similar. Various demographic factors, clinical assessments, and laboratory tests were examined in relation to greater severity. When comparing the two periods, we observed predominance of the Wild virus during the first wave and the Gamma variant during the second, with no significant difference in outcomes. The findings suggest that despite the association of many factors with increased severity, the temporal variation between the two periods did not result in a notable divergence in the measured outcomes. The COVID-19 pandemic has lasted for a long time, with periods marked by peaks of cases, often caused by the emergence of viral variants, resulting in higher infection rates and rapid dissemination but, for variant Gamma, no apparent greater virulence.

**Funding:** The authors received no specific funding for this work.

**Competing interests:** The authors have declared that no competing interests exist.

## Background

The COVID-19 pandemic caused by the new coronavirus (*Severe Acute Respiratory Syndrome Coronavirus 2*—SARS-CoV-2) has been one of this century's most significant global health challenges. It was responsible for approximately 760 million cases and more than 6.8 million deaths worldwide until March 2023 [1]. With the United States leading the number of cases and Brazil ranking sixth, the American continent is the second most affected by the pandemic [1].

Similar to other countries, Brazil has had fluctuations in the number of cases and fatalities since the first documented COVID-19 case in February 2020. Although the pandemic has reached virtually every country, it has not occurred equally worldwide. In some countries, the waves were shorter and more intense than in other countries. For instance, Brazil was still in the second wave, whereas Iran was affected by four waves until March 2021 [2]. The population's susceptibility, restriction measures, health services responses, and SARS-CoV-2 transmission rate were influential in determining the occurrence and severity of the waves [3].

Late in 2020, Brazil experienced a new rise in hospital admissions. Concurrently, the description of new variants of interest (VoI) and variants of concern (VoC) raised the hypothesis that a more contagious or virulent variant could explain the increase in number of cases [4]. The B.1.195 SARS-CoV-2 strain was introduced in Brazil at the beginning of the pandemic and was soon replaced by variant B.1.1.28 [5]. Variant of concern Gamma was first detected in four Japanese travellers who returned from Brazil in early 2021, [6] and became more prevalent as the second wave progressed. When the Gamma variant first emerged in northern Brazil, it was associated with mutations that increased viral load and transmissibility, and it took less than three months for it to become dominant [4].

Any individual can be affected by COVID-19, and up to 20% of those who are symptomatic may progress to severe disease, with older adults and those with chronic conditions being particularly susceptible to experiencing severe and critical illness [7–9]. As a result, they were the most prevalent group in terms of hospitalizations during the first wave [8–10] In the second wave, an increase in hospitalization among young and not chronically ill individuals raised the hypothesis of a shift in the characteristics of patients requiring hospitalization [11]. Thus, the present study aimed to examine the clinical and epidemiological characteristics of individuals hospitalized with COVID-19 during two pandemic periods caused by distinct VOC of the SARS-CoV-2 in a tertiary referral hospital in southern Brazil.

## Material and methods

### Study design

A retrospective cohort study was conducted at Complexo Hospital de Clínicas/Universidade Federal do Paraná (CHC-UFPR), a tertiary academic hospital in Paraná, southern Brazil. The Human Research Ethics Committee of the Complexo Hospital de Clínicas da Universidade Federal do Paraná approved the study (#n 51400121.9.0000.0096). The data collection was conducted after obtaining ethical approval from the committee on September 14, 2021, continuing until August 31, 2022. Subsequent to data inclusion in the database, all participants were assigned unique numerical identifiers to ensure their complete anonymity and to safeguard their identities from all authors involved. Due to its retrospective nature, the informed consent was waived.

The study was conducted with a convenience sampling of patients hospitalized in respiratory units at CHC-UFPR suspected of COVID-19 disease and patients in other critical units identified with B34.2 ICD 10 (infection by SARS-CoV-2 of unspecified location) between March 11[th], 2020 and August 1[st], 2021. The inclusion criteria were patients with confirmed or

probable COVID-19 disease. The exclusion criteria were patients under the age of 18, those who tested positive for SARS-CoV-2 infection due to elective inpatient care, those whose symptoms were attributed to other causes, and patients with missing data.

Clinical and epidemiological characteristics of patients hospitalized due to COVID-19 at two different epidemiological moments of the pandemic were compared. The first moment comprised the period between March 11[th], 2020 and February 16[th], 2021, and the second comprised the period between February 17[th], 2021 and August 1[st], 2021. The cutoff date was chosen because the first COVID-19 infection in Curitiba by the Gamma variant was confirmed on February 17[th], 2021.

Subsequently, clinical and epidemiological characteristics were assessed to identify risk factors associated with the progression to critical illness and trace survival curve comparisons between the first and second periods.

## Data collection

Epidemiological, clinical, and outcome data were collected based on a standard notification form for severe acute respiratory syndrome (SARS) from the Influenza Epidemiological Surveillance Information System (SIVEP Influenza), a national database used to monitor cases of respiratory infections. Chest computed tomography (CT) data and other relevant clinical information were obtained through the Hospital's Computerized System.

## Definitions

The COVID-19 case was classified as confirmed, probable, or discarded, as presented in Table 1.

The presence of two symptoms of fever, cough, dyspnea, and desaturation was considered clinically compatible with COVID-19. Findings consistent with SARS-CoV-2 infection identified on chest CT included ground-glass consolidations, predominantly peripheral, and involvement of multiple lobes. All CT scans were analyzed and interpreted by a radiologist, and the request for the exam was the assistant physician's decision. For the evaluation, only the first laboratory exams performed at the patient's admission (or, in the case of nosocomial infection, those closest to the date of onset of symptoms) were considered.

A Clinical Progression Scale was used to classify the disease severity. The score ranges from 0 to 10, and the parameters include level of medical assistance (outpatient, ward, or intensive care unit) and oxygen supplementation (no need for oxygen supplementation, nasal catheter supplementation, high-flow nasal cannula, non-invasive ventilation, or orotracheal intubation). Scores 4 and 5 were classified as moderate, and scores above 6 were classified as severe

**Table 1. COVID-19 classification criteria.**

| Classification | Criteria |
|---|---|
| Confirmed case | • SARS-CoV-2 RT-PCR positive or,<br>• SARS-CoV-2 antigen test positive or,<br>• IgM reagent for SARS-CoV-2 and chest computed tomography (CT) with typical COVID-19 findings and COVID-19 clinical features. |
| Probable case | • IgM reagent for SARS-CoV-2 and chest computed tomography (CT) or,<br>• IgM reagent for SARS-CoV-2 and COVID-19 clinical features. |
| Discarded | Do not meet any of the above-mentioned criteria. |

RT-PCR = Reverse Transcription Polymerase Chain Reaction

disease [12]. The Charlson Comorbidity Index (CCI) was used to categorize the studied sample according to the comorbidities [13].

## Virus genotyping

Virus genotyping was performed on clinical samples from patients included in the study who had a quantification cycle (Cq) value <35 on the SARS-COV-2 RT-qPCR diagnostic.

The viral RNA was isolated from the clinical sample using the Extracta 32 automated extraction system with the EXTRACTA DNA and viral RNA kit (Loccus, SP, Brazil) according to the manufacturer's guidelines.

We used two probe-based genotyping systems to characterize SARS-CoV-2 variants. The first identified Alpha, Beta or Gamma, and Wild type. The second one determined the Delta variant and differentiated Beta from Wild-type, Gamma, and Zeta. RT-qPCR reaction was performed in the GoTaq[TM] Probe 1-step RT-PCR system (Promega Bio Sciences, LLC. San Luis Obispo, CA, USA) according to the manufacturer's guidelines, with an input of 2.5uL RNA in a total reaction volume of 10uL [14].

## Data analysis

Statistical analyses were performed using the R Studio software, version 3.6.1. A descriptive analysis showed the clinical, epidemiological, laboratory, and molecular features. Data from quantitative variables were presented as medians and interquartile range (IQR). Univariate analysis was performed using Fisher's exact and Chi-squared tests for categorical variables. Mann-Whitney and Kruskal-Wallis rank sum tests with Tukey's multiple post hoc comparisons were used for continuous variables, as appropriate. We used Kaplan-Meier analysis to estimate the probability of survival and log-rank testing for between-group comparison. Receiver operating characteristic (ROC) curves were built to evaluate laboratory tests to find a cutoff point for poor outcomes. To evaluate risk factors associated with the outcomes, adjusted OR (aOR) was calculated using the multivariate model with a stepwise selection of variables, with a cut-off point of p<0.2. All statistical tests were two-sided, with significance set at p< 0.05. A confidence interval (CI) of 95% was used to adjust the estimates.

## Results

A total of 4,546 individuals over the age of 18 were hospitalized in COVID-19 settings from March 11[th], 2020 to August 1[st] 2021 (Fig 1). The first wave was from March 11[th], 2020 to February 16[th], 2021 (a total of 343 days), and the second wave from February 17[th], 2021 to August 1[st] 2021 (165 days).

Fig 2 shows the dynamics in the number of cases for the city of Curitiba, PR, (Brazil), the hospitalizations at CHC-UFPR, and the viral variant detection in the region over the same time period. As observed, Wild virus and Gamma variant predominated in the first and second studied periods, respectively.

### Clinical and demographic characteristics of the study population

The overall median age was 56 years (IQR 44–66), with a predominance of men (53.6%) and white race (89.6%). Overall, 2,359 (81.7%) patients reported having prior diseases, the median Charlson Comorbidity Index score was 2 points (IQR 1–4). Cardiovascular diseases were the most common comorbidities (46.1%), followed by type 2 diabetes (25.8%) and obesity (17.9%). The median time of symptom onset was 9 days (IQR, 6–12) before hospitalization. The median time of hospitalization and mechanical ventilation were 7 (IQR 4–16) and 8 (IQR

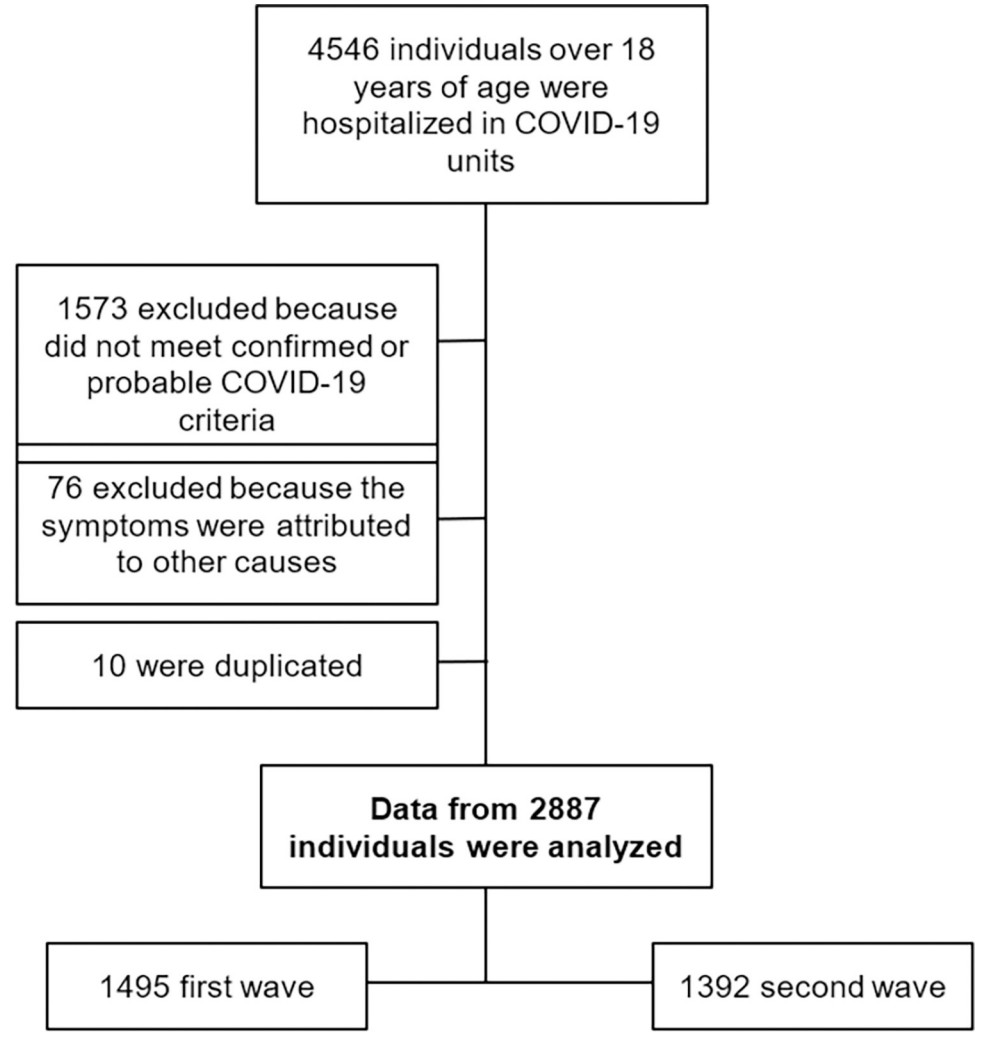

**Fig 1. Flow-chart of study design.**

4–17) days, respectively. Regarding disease severity, 1,109 (38.4%) individuals had severe disease, while 1,093 (37.9%) were admitted to critical units; 641 (22.2%) patients died during hospitalization, while 77.8% were discharged.

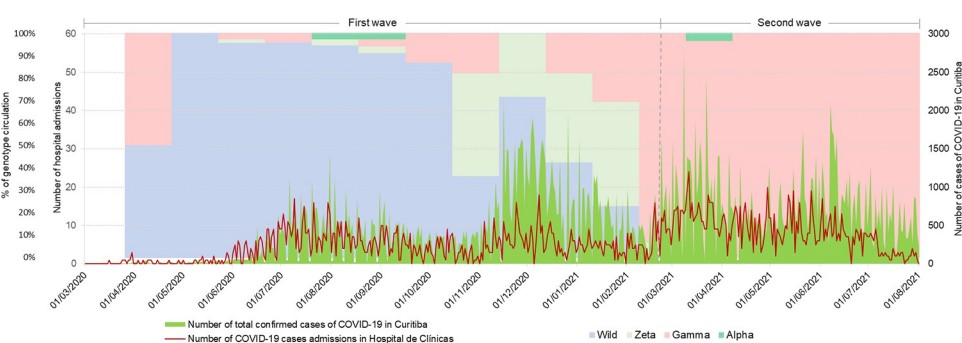

**Fig 2. Temporal distribution of COVID-19 cases in Curitiba (Brazil), hospital admissions, and viral genotype.**

## Comparison of clinical and demographic data between the first and second waves

Comparative analysis showed a predominance of men in both waves, 54% in the first and 53.2% in the second. There was a significant reduction in median age in the second wave (52 years, IQR 42–63) compared to the first (59 years, IQR 47–68 years), p<0.001. The Wild strain was more prevalent in the first wave (78%), while Gamma variant was more prevalent in the second wave (95.3%).

Table 2 summarizes the clinical and demographic characteristics of participants included in the study, divided by pandemic waves. The frequency of comorbidities was similar in both periods. In the second wave, there was a lower prevalence of cardiovascular disease (42.2% versus 49.7% in the first wave, p<0.001) and type 2 Diabetes Mellitus (22.4% versus 28.9% in the first wave, p<0.001). There were more hospitalizations among obese individuals in the second wave compared to the first (23% versus 13.2%, respectively; p<0.001). Further, the Charlson

**Table 2. Baseline characteristics by pandemic wave of individuals hospitalized at Complexo Hospital de Clínicas UFPR, from March 11th, 2020 to August 1st, 2021.**

| | First pandemic wave n (%) | Second pandemic wave n (%) | p value |
|---|---|---|---|
| **Median age (in years)** | 59 (IQR 47–68) | 52 (IQR 42–63) | <0.001 |
| **Sex** | | | 0.6602 |
| Female | 687 (46) | 652 (46.8) | |
| Male | 808 (54) | 740 (53.2) | |
| **Race (n)** | | | 0.0022 |
| White | 1285 (87.9) | 1264 (90.8) | |
| Non-white | 177 (12.1) | 118 (8.7) | |
| **Genotype (n)** | **369** | **169** | <0.001 |
| Wild | 288 (78.1) | 2 (1.2) | |
| Zeta | 52 (14.1) | 5 (2.9) | |
| Gamma | 27 (7.3) | 161 (95.3) | |
| Alpha | 2 (0.5) | 1 (0.6) | |
| **Comorbidities** | | | |
| Cardiovascular diseases | 743 (49.7) | 588 (42.2) | <0.001 |
| Hematological diseases | 27 (1.8) | 21 (1.5) | 0.632 |
| Liver diseases | 37 (2.5) | 26 (1.9) | 0.323 |
| Asthma | 59 (3.9) | 60 (4.3) | 0.691 |
| Type-2 diabetes | 432 (28.9) | 312 (22.4) | <0.001 |
| Neurological diseases | 88 (5.9) | 66 (4.7) | 0.199 |
| Lung diseases | 116 (7.8) | 54 (3.9) | <0.001 |
| Immune diseases | 52 (3.5) | 32 (2.3) | 0.076 |
| Kidney diseases | 73 (4.9) | 41 (2.9) | 0.011 |
| Obesity | 197 (13.2) | 320 (23) | <0.001 |
| **Respiratory support** | | | <0.001 |
| None | 117 (7.8) | 66 (4.7) | |
| Non invasive | 820 (54.8) | 840 (60.3) | |
| Invasive | 558 (37.3) | 486 (34.9) | |
| **Disease severity** | | | 0.658 |
| Mild | 927 (62) | 851 (61.1) | |
| Severe | 568 (38) | 541 (38.9) | |
| **Outcome** | | | 0.564 |
| Discharge | 1170 (78.3) | 1076 (77.3) | |
| Death | 325 (21.7) | 316 (22.7) | |

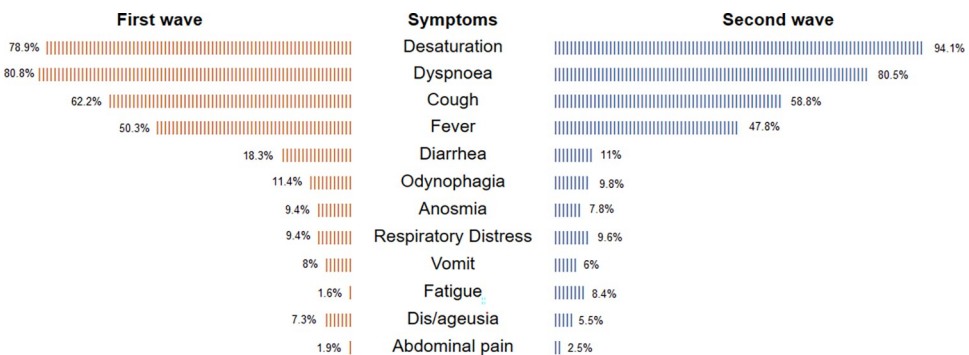

**Fig 3. Comparison between the frequency of the reported symptoms at the time of hospital admission during the first and second pandemic waves.**

Comorbidity Index median score in the first wave was 3 (IQR 1–4), while in the second wave was 2 (IQR 0–4), p<0.001.

There was no significant difference in the frequency of reported symptoms between the waves, and the most common symptoms were desaturation (78.9% in the first wave and 94.1% in the second), dyspnea (80.8% and 80.5%), and cough (62.2% and 58.8%), as shown in Fig 3.

In the first pandemic wave, the median time from the onset of symptoms to hospital admission was 8 days (IQR 5–11), while in the second wave, it was 10 days (IQR 8-12); p<0.001. The length of hospitalization was similar, 8 days (IQR 4–16) and 7 days (IQR 4-16) in the first and second waves, respectively; p = 0.684. The length of intensive care unit (ICU) stay was 9 days (IQR 4–18) in the first wave and 10 days (IQR 4–21) in the second wave, p = 0.257, and the median time on mechanical ventilation was 7 (IQR 3–15) and 10 (IQR 5-21) days in the first and second waves, respectively, p<0.001. The median time from the onset of symptoms to orotracheal intubation was 8 (IQR 5–12) and 10 (IQR 8–13) days in the first and second waves, respectively, p<0.001. The proportion of patients that did not require oxygen support was higher in the first (7.8%) than in the second wave (4.7%); p<0.001. Regarding disease severity, there was no difference between the two pandemic waves, and the proportion of patients admitted to the ICU was 40.3% in the first wave and 35.3% in the second. Similarly, no significant difference was observed in in-hospital fatality rate in both waves (21.7% in the first wave versus 22.7 in the second; p = 0.5642). In the second wave, there was an increase in hospitalizations of individuals aged 31 to 40 and 51 to 60 years with severe illness, both with p < 0.05 when comparing the severity of the disease across age groups. Additionally, during the first wave, a higher severity of illness was observed among age groups between 61 and 70 years, between 71 and 80 years, and those aged over 80 years, all with p < 0.05.

Overall, 78.6% of the laboratory samples were collected within 48 hours after admission. Patients admitted during the second wave had significantly higher marker inflammatory test values, such as lactate dehydrogenase, ferritin, C-reactive protein, procalcitonin and D-dimer. Table 3 summarizes the laboratory results.

## Disease severity analysis

Univariate analysis of factors related to disease severity shows a higher median age in patients with severe disease compared to those with moderate disease (59 versus 53 years, respectively; p<0.001). Male sex was also significatively related to severe disease, p = 0.004. The Charlson Comorbidity Index score was higher in the severe disease group, with 3 points versus 2 points in the moderate disease group, p<0.001. Hospitalization length was longer in patients with

**Table 3. Laboratory test results of individuals hospitalized at CHC-UFPR from March 11th, 2020 to August 1st, 2021.**

| Laboratory results | First pandemic wave Median (IQR) | Second pandemic wave Median (IQR) | p value |
|---|---|---|---|
| Lymphocytes count (NR 0,8–4,9x10³/uL) | 1310.29 (569.25–1326) | 980.59 (519.5–1198.5) | <0.001 |
| Total bilirubin (NR <1.2mg/dL) | 0.44 (0.33–0.64) | 0.46 (0.32–0.64) | 0.8884 |
| Creatinine (NR 0.72–1.25mg/dL) | 0.86 (0.74–1.18) | 0.80 (0.70–1.04) | <0.001 |
| Aspartate aminotransferase (NR 5–34 U/L) | 37 (25–57) | 41 (28–65) | <0.001 |
| Alanine aminotransferase (NR 0–55 U/L) | 39 (23–63) | 44 (28–75) | <0.001 |
| DHL (NR 125–220 U/L) | 383 (289–503) | 423 (328–559.5) | <0.001 |
| Ferritin (NR 4.63–204 ng/mL) | 1079.26 (478–2023.49) | 1246.36 (620.34–2487.88) | <0.001 |
| CRP (NR ≤0.5 mg/dL) | 7.29 (3.26–13.73) | 9.51 (4.67–16) | <0.001 |
| Procalcitonin (NR<0.5 ng/mL) | 0.21 (0.08–0.72) | 0.16 (0.07–0.6) | 0.128 |
| D-dimer (NR <0.55 mg/L) | 1.06 (0.56–2.4) | 1.06 (0.59–2.61) | 0.453 |

NR = normal range. CRP = C-reactive protein. DHL = lactate dehydrogenase

severe disease (15 days; IQR 9–26) compared to those with moderate disease (5 days; IQR 3–9), p<0.001. Similar findings were observed for the length of ICU stay (10 days [IQR 4–20] in severe disease versus 4 days [IQR 2–6] in moderate disease; p<0.001). Higher values of ferritin (median 1549ng/mL [IQR 682.5–2885.75ng/ml]; p<0.001) and c-reactive protein (CRP) (median 12.2mg/dL [IQR 6.46-16ng/ml]; p<0.001) were also associated with severe disease.

Table 4 shows the multivariate analysis of factors associated with disease severity. Age, sex, and obesity remained independent risk factors for severe disease.

## Outcome analysis

Patients discharged were significantly younger and more likely to be female compared to those who died (median 53 years versus 65 years, p<0.001; and p = 0.0032 for sex), but no differences were observed for race. Mortality was more frequent among patients with higher Charlson Comorbidity Index scores, and among those with kidney and cardiovascular diseases. Although obesity was a risk factor for disease severity, no significant difference was observed in the fatality rate. Also, no differences in outcomes were found when the genotypes were compared. Furthermore, patients who died were admitted significantly earlier to the hospital, and the median time between hospital admission and the outcome was longer.

In the multivariate analysis, age, male sex (OR 1.3, CI 1.03–1.63, p = 0.024), disease severity classification (OR 5.04, CI 2.19–10.52, p<0.001) and need of invasive ventilatory support (OR 5.64, CI 2.81–12.65, p<0.001) remained as independent risk factors for death.

**Table 4. Multivariate analysis of predictors associated with disease severity.**

| Variables | Odds ratio | CI (95%) | p value |
|---|---|---|---|
| Age | 1.01 | 1–1.02 | 0.044 |
| Male | 1.23 | 1.03–1.46 | 0.021 |
| Charlson Comorbidity index score | 1.11 | 1.03–1.2 | 0.005 |
| Fever | 1.22 | 1.03–1.46 | 0.025 |
| Neurologic disease | 1.68 | 1.09–2.6 | 0.020 |
| Obesity | 2.14 | 1.71–2.68 | <0.001 |
| Length of hospital stay | 1.10 | 1.09–1.11 | <0.001 |

Abbreviations: CI, confidence interval.

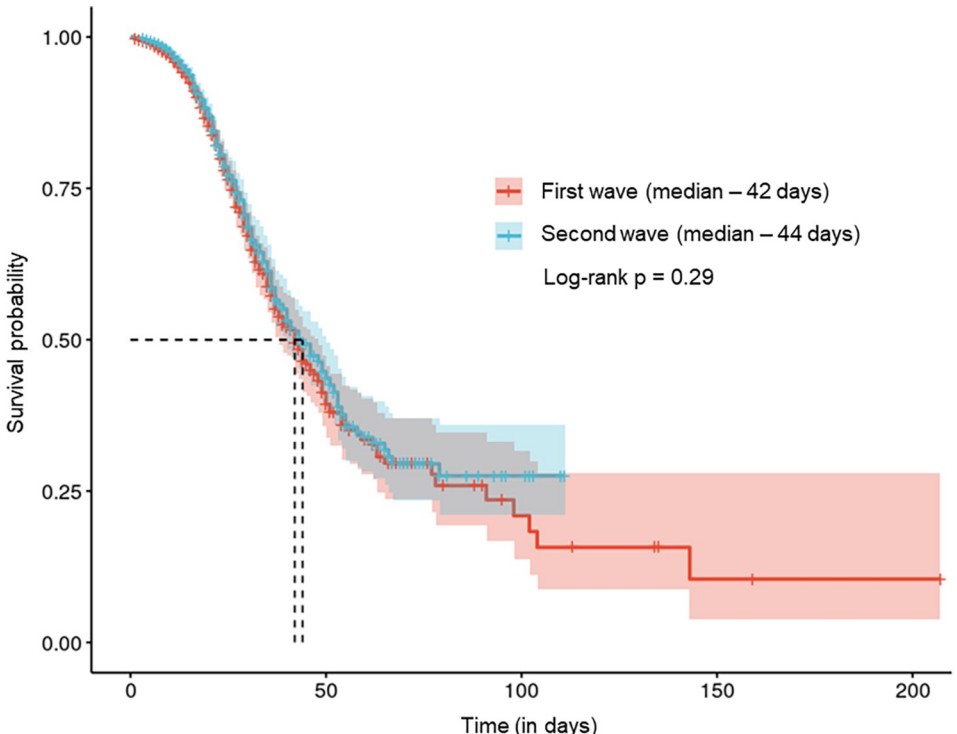

**Fig 4. Kaplan-Meier survival analysis of patients admitted at CHC-UFPR comparing the first and second pandemic waves.**

Comparing the laboratory tests, patients who died had more pronounced lymphopenia and higher values of lactate dehydrogenase (DHL), ferritin, CRP, and d-dimer compared to those who survived. Receiver operating characteristic (ROC) curves were built to analyze laboratory tests in order to determine a cutoff point for poor outcomes. A value of 10.28mg/L for CRP revealed a sensitivity and specificity for worst outcomes of 61.3% and 62.9%, respectively (area under the curve [AUC] = 0.655, 95% CI 0.628–0.681). Creatinine exhibited a statistically significant difference in the outcome despite being somewhat over the reference value, with a median of 1.92mg/dL for patients who died. Moreover, values above 1.05mg/dL demonstrated 80.1% specificity but only 57.1% sensitivity for adverse outcomes (AUC = 0.717, 95% CI 0.690–0.745). Patients ages ≥ 58.5 years, Charlson Comorbidity Index score ≥ 2.5, ferritin levels ≥ 369.35ng/mL, DHL level ≥ 432.5U/L, and D-dimer levels ≥ 1.25mg/L were related to poor outcomes (AUC values of 0.708, 0.696, 0.613, 0.685, and 0.678, respectively) (supplementary material).

Fig 4 shows a comparison of survival rates for the first and second waves, and no significant differences were found.

At the end of the year 2020, there was an increase in the number of cases. It occurred at a point in time when the learning curve for managing the disease had already become better established, alongside the implementation of corticosteroid usage protocols. Taking this into consideration, a comprehensive sub-analysis was conducted to compare the outcomes between two distinct periods: November 1st, 2020, to February 16th, 2021, and February 17th, 2021, to August 1st, 2021. Despite the temporal shift, the results revealed that there was no statistically significant difference in outcomes between these two timeframes (p = 0.2175). This suggests that the observed results are consistent across both periods, and the temporal variation did not have a substantial impact on the measured outcomes.

## Discussion

The COVID-19 pandemic has had a profound impact on health services and also on socio-economic aspects worldwide. It is a novel disease caused by an emerging pathogen and has been present for approximately three years. During this period, we faced pandemic waves caused by the emergence of viral variants as a result of persistent viral replication independent of natural, passive, or active immunity. In Brazil, between the first and second pandemic waves, in which Wild virus and Gamma variant prevailed, respectively, frontline health care professionals reported perceiving more severe cases and excessive work overload in the second wave. In this study, to evaluate the impact of these two events, we analyzed data from a single tertiary facility to compare the hospitalizations between the first two waves of the COVID-19 pandemic.

Overall, a significant difference was observed in patients' age, who were younger in the second wave compared to the first wave. Despite chronic diseases being more prevalent in the first wave, obesity was significantly more prevalent in the second wave. However, no significant difference was observed in the fatality rate, which was approximately 22% [8, 10, 15]. Other studies also found a decrease in fatality rate throughout the pandemic [2].

The first reports describing the clinical and epidemiological characteristics of COVID-19 showed high hospitalization rates among men and older adults [9, 16]. Subsequent studies showed that advanced age and male sex were associated with more severe disease and the potential need for mechanical ventilation and admission to critical units, regardless of pre-existing comorbidities [17]. A Mexican study showed that being above the age over 65 was associated with a higher risk of hospitalization, especially when obesity was present, [18] and a study from India showed a significant association between advanced age and the risk of progressing to severe and critical disease [19]. Immunosenescence is responsible for changes in the innate and adaptive immune response, and advanced age is naturally related to poor immune responses. Differences in the immune system, sex hormones, physiological factors, and lifestyle have been linked to a higher risk of poor outcomes in men [20].

Regarding comorbidities, a Brazilian population-based study showed that 16% of individuals did not have comorbidities, [8] while an observational study from the United States showed that 84,7% of the patients had at least one comorbidity at admission [21]. Since the median age found in our study was lower than previously reported in the literature and since older people tend to have more comorbidities, these may explain why our study found more patients without comorbidities. Similar to what was previously reported, the median time from symptom onset to hospitalization varied between 6 and 8 days [8, 15].

The overall fatality rate found in this study was considerably lower than those reported in other studies, which can reach over 40% [22]. A national study conducted in Brazil compared the characteristics of hospitalizations during the pandemic's early months across the country's five macroregions and found a nationwide fatality rate of 38% and 31% in the Southern region [8].Lower fatality rates were observed in studies conducted at later phases of the pandemic, [15] which may suggest a positive learning trend and more readily available physical and professional infrastructure. Adopting health care protocols based on scientific evidence, such as using a high-flow nasal cannula in hospital wards, [23] using corticosteroids and a qualified multidisciplinary health care team contributed to the better results observed in this study.

Similar to the findings of an African population-based study, this comparative analysis showed a rise in the median time from the symptom onset to hospital admission in the second wave compared to the first, showing a bed shortage and a healthcare system overload [24]. This hypothesis may be supported by the increase in the median time of mechanical ventilation usage in the second wave without an increase in the length of ICU stay, suggesting a higher turnover of ICU beds to fulfill the demand for this level of care. Furthermore, the lack

of beds in critical units may have delayed patients´ intubation at the appropriate moment, resulting in more severe cases being intubated and extending the duration of mechanical ventilation.

As shown in previous reports, there was a reduction in the median age of hospitalized individuals in the second wave compared to the first wave [11]. A Mexican study compared the first three waves of the pandemic and showed an increase in the risk of hospitalization among young patients throughout the waves, with a significant higher OR in the age group of 25–29 years in the third wave compared to the age group above 45 years in the first wave [2, 20]. The median age of hospital admissions was reduced by nearly ten years, and there was a 7% increase in hospital admissions among individuals aged 0 to 19 [18, 20]. Some hypotheses, such as previous infection immunity and vaccination coverage, may explain this finding. In Curitiba, the COVID-19 vaccination began on January, 2021, for older adults, reaching the age group of 65 years in mid-April 2021. The role of the Gamma variant is also questioned, but a causal relationship cannot be established. It is possible that, as the variant becomes more transmissible, there has been an increase in cases among the population that has remained vulnerable to infection, such as the economically active population.

The proportion of patients with comorbidities in the literature varies by wave and region. Our analysis found an increase in hospitalizations of individuals without comorbidities during the second wave, which is in agreement with an Indian study, but differs from a Spanish cohort that found an increase in ICU admissions among individuals with comorbidities [19–22, 24, 25]. These contradicting findings might be attributed to socioeconomic factors and challenges with healthcare access, as developing countries face more significant barriers to health care and possibly less diagnosis and control of chronic diseases. The prevalence of Type 2 Diabetes Mellitus and cardiovascular diseases have both decreased significantly, while the prevalence of obesity has increased and has been associated with more severe disease, but not with mortality.

The proportion of patients who did not require oxygen support was lower in the second wave. These findings should be interpreted with caution, since oximeters for monitoring saturation at home were made available as part of the restructuring of the primary healthcare system. Improved assistance in these health units occurred, as did the provision of home oxygen therapy. Furthermore, during the second wave, a more pronounced scarcity of hospital beds was observed, leading to the prioritization of hospitalization for patients with heightened disease severity.

In this study, laboratory tests revealed increased inflammatory markers such as CRP, DHL, and ferritin levels in both waves, but they were significantly higher in the second one. Changes in D-dimer levels and lymphopenia were also found. These findings are consistent with previous studies, [20, 21] which found that increasing CRP levels by one unit raises the probability of developing severe disease by 0.06%. CRP is more than a marker of inflammation; it is also capable of maintaining the inflammatory response by attracting leukocytes to areas of inflammation and causing the release of pro-inflammatory cytokines. It is reasonable to assume that high levels of CRP are associated with elevated interleukin 6 (IL-6) levels, given that the CRP gene is predominantly activated by IL-6 [20]. Elevated serum levels of LDH, a protein present in the cell cytoplasm, may suggest tissue injury or necrosis. According to previous studies, having high levels of LDH may increase the risk of death by approximately four times [17].

In contrast to the second wave's subjective perceptions, [11] there was no increase in the severity of hospitalized cases or the number of deaths. The two waves had no statistically significant difference in disease severity or outcome, and the survival curves were similar [11, 24–26]. Several factors might explain these findings, including an increase in the number of beds, the use of high-flow nasal cannulas, and the learning curve in disease management, which was associated with a standardized care protocol, enabling an optimized clinical management

throughout the pandemic, resulting in fatality rates lower than previously reported in the country.

The first months of the pandemic in Brazil were primarily caused by two strains, B.1.1.28 and B.1.133. The Gamma variant replaced the previous lineages in early 2021 and became the main variant in less than three months, [27] as this study also showed. Unlike the first wave, in which the variant spread primarily from densely populated urban areas, the second wave originated in the Amazon, a geographically remote area with difficult access. The Gamma variant was estimated to have an effective higher median reproductive number (Re) and was 1.56–3.06 times more transmissible than previous non-Gamma variants [27]. While earlier research has indicated elevated likelihoods of hospitalization and admission to ICU among Gamma cases [3, 5, 28–31], it is noteworthy that even though Gamma cases were predominant during the second wave, we did not observe corresponding higher rates of ICU admissions or fatalities in our study. Other studies also found differences in genotype prevalence across waves, including a decrease in fatality rate following the emergence of VOC Alpha [15]. Although the Gamma variant has spread to many countries, it has been identified mainly on the American continent (with 58% of the cases in Brazil and 26% in the USA); [30] therefore, few studies have compared the impact of its introduction in clinical and epidemiological aspects outside of Brazil.

In the univariate analysis, age, male sex, fever, neurologic disease, Charlson Comorbidity Index score, obesity, and length of hospital stay were all associated with higher disease severity. These factors remained independent risk factors after the multivariate analysis, as shown in previous studies [2, 17, 19, 21, 22].

This study has limitations, mainly due to its retrospective nature, which depends on accurate and comprehensive medical record entries. Moreover, the dynamic evolution of disease understanding and treatment, alongside fluctuations in bed availability to fulfill the demand for hospitalizations, may have an influence on certain outcomes. Additionally, we were unable to conduct a genetic analysis on the entire sample.

In conclusion, the COVID-19 pandemic has lasted for a prolonged period, marked by oscillations in cases termed waves. These fluctuations are frequently linked to the emergence of viral variants harboring mutations that enhance viral fitness, replicative capacity, and immune evasion, leading to increased infection rates and rapid spread. Nevertheless, Gamma VoC did not display any apparent greater virulence compared to the Wild strain, and the disease severity remained comparable to previous variants.

## Supporting information

**S1 Fig. ROC of laboratory parameters–Creatinine, Lactate dehydrogenase, Ferritin, CRP and D-Dimer—in predicting the in-hospital mortality of COVID-19 patients.** Note: ROC, receiver operating characteristic; AUC, area under the ROC curve; CRP, C-reactive protein. (TIF)

**S1 Data.**
(XLSX)

## Acknowledgments

The authors would like to sincerely thank the Virology Laboratory of the Hospital de Clínicas at the Federal University of Paraná, the Genetics Department Laboratory of the Federal University of Paraná, and the Respiratory Virus and Measles Laboratory of Fiocruz/RJ for their invaluable support, insightful information exchanges, and significant contributions to this research. Your collaboration has greatly enriched the quality and depth of our work.

## Author Contributions

**Conceptualization:** Sonia M. Raboni.

**Data curation:** Felipe Zhen, Guilherme Yamaguto, Emanueli C. S. Costa, Vitor L. Moriya, Ricardo Petterle.

**Formal analysis:** Felipe Zhen, Guilherme Yamaguto, Emanueli C. S. Costa, Vitor L. Moriya, Ricardo Petterle, Sonia M. Raboni.

**Funding acquisition:** Sonia M. Raboni.

**Investigation:** Natalia R. Domino, Bruna A. Lapinscki.

**Methodology:** Natalia R. Domino, Bruna A. Lapinscki, Luciane A. Pereira, Meri B. Nogueira.

**Project administration:** Sonia M. Raboni.

**Resources:** Sonia M. Raboni.

**Supervision:** Sonia M. Raboni.

**Validation:** Natalia R. Domino, Bruna A. Lapinscki, Sonia M. Raboni.

**Visualization:** Natalia R. Domino, Bruna A. Lapinscki, Luciane A. Pereira, Meri B. Nogueira.

**Writing – original draft:** Natalia R. Domino.

**Writing – review & editing:** Sonia M. Raboni.

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
