## [Decision Letter · Decision Letter 0]

26 Sep 2023

PONE-D-23-26555Assessing the impact of the Gamma variant on COVID-19 Patient admissions in a Southern Brazilian tertiary hospital - A comparison of dual pandemic phasesPLOS ONE

Dear Dr. Raboni,

Thank you for submitting your manuscript to PLOS ONE. After careful consideration, we feel that it has merit but does not fully meet PLOS ONE’s publication criteria as it currently stands. Therefore, we invite you to submit a revised version of the manuscript that addresses the points raised during the review process.

ACADEMIC EDITOR: Kindly attend to all comments and corrections raised by the reviewers. Ensure to provide detailed response accordingly.

We look forward to receiving your revised manuscript.

Kind regards,

Olatunji Matthew Kolawole, Ph.D.

Academic Editor

PLOS ONE

Journal Requirements:

 "No - The funders had no role in study design, data collection and analysis, decision to publish, or preparation of the manuscript."

5. Please ensure that you include a title page within your main document. You should list all authors and all affiliations as per our author instructions and clearly indicate the corresponding author.

6. Please upload a copy of Figure 1, to which you refer in your text on page 6. If the figure is no longer to be included as part of the submission please remove all reference to it within the text.

Additional Editor Comments:

A well written and detailed article. The data presented are timely and very important information for the collective global efforts against SARS-CoV 2 and COVID-19.

Authors are to keenly attend to the comments raised by reviewers, and provide responses accordingly.

Reviewers' comments:

Reviewer's Responses to Questions

**Comments to the Author**

1. Is the manuscript technically sound, and do the data support the conclusions?

Reviewer #1: Yes

Reviewer #2: Yes

Reviewer #3: Yes

2. Has the statistical analysis been performed appropriately and rigorously? 

Reviewer #1: Yes

Reviewer #2: I Don't Know

Reviewer #3: Yes

3. Have the authors made all data underlying the findings in their manuscript fully available?

Reviewer #1: Yes

Reviewer #2: Yes

Reviewer #3: Yes

4. Is the manuscript presented in an intelligible fashion and written in standard English?

Reviewer #1: Yes

Reviewer #2: Yes

Reviewer #3: Yes

5. Review Comments to the Author

Reviewer #1: It will be nice, if the authors can have access to the vaccination status of these patients including the vaccine types and how many shots and compare these with the patient outcome. The paper is really valuable and acceptable for publication.

Reviewer #2: ABSTRACT: In line 8, it is not clear what CHC-UFPR is.

BACKGROUND: Lines 49-50 should be referenced.

RESULTS: What are these higher inflammatory test values?

DISCUSSION: Line 292 would better read--patients ages.

Reviewer #3: I have not concerns concerning the manuscript. I like the fact that datas gotten from the Research project are available. Though I would have appreciated that the Research is extended beyond Brazil.

6. PLOS authors have the option to publish the peer review history of their article (what does this mean?). If published, this will include your full peer review and any attached files.

Reviewer #1: **Yes: **Ado Garba Abubakar

Reviewer #2: No

Reviewer #3: **Yes: **Emmanuel Adamolekun

---

## [Author Response · Author response to Decision Letter 0]

6 Nov 2023

Editor #:

The manuscript has been reviewed and adjusted to meet PLOS ONE's style requirements.

2.Thank you for stating the following financial disclosure:

 "No - The funders had no role in study design, data collection and analysis, decision to publish, or preparation of the manuscript."

The authors received no specific funding for this work. The research was conducted without any external financial support, and the authors were solely responsible for the study design, data collection, analysis, decision to publish, and manuscript preparation. As requested, the amended statements was included in the cover letter. 

3. Upon re-submitting your revised manuscript, please upload your study’s minimal underlying data set as either Supporting Information files or to a stable, public repository and include the relevant URLs, DOIs, or accession numbers within your revised cover letter. 

As requested, an anonymized spreadsheet containing the demographic, clinical, and laboratory data of research participants will be made available as Supporting Information file. 

Please check the answer to query 3.

5. Please ensure that you include a title page within your main document. You should list all authors and all affiliations as per our author instructions and clearly indicate the corresponding author. 

A title page has been included into the main document, listing all authors and affiliations and indicating the corresponding author.

6. Please upload a copy of Figure 1, to which you refer in your text on page 6

A copy of Figure 1 was uploaded, which is referenced in the text on page 8 of the updated manuscript. The figure is now included as part of the submission.

The reference list has been thoroughly examined and updated. None of the references was retracted.

8. While revising your submission, please upload your figure files to the Preflight Analysis and Conversion Engine (PACE) digital diagnostic tool.

The figure files have been uploaded to the Preflight Analysis and Conversion Engine (PACE) digital diagnostic tool to meet PLOS One figure specifications. The figures are provided in TIFF format, with a resolution of 300 ppi, and each file does not exceed 10MB in size, as per the requested requirements.

Reviewer #1: 

1) It will be nice, if the authors can have access to the vaccination status of these patients including the vaccine types and how many shots and compare these with the patient outcome. The paper is really valuable and acceptable for publication.

The vaccination status of patients was not analyzed in this study due to the presence of data from periods without vaccination and extended periods with only one vaccine dose. Exploring the true impact of the vaccine in this situation proved challenging. Another study conducted at our institution specifically aims to address this analysis.

Reviewer #2:

ABSTRACT: In line 8, it is not clear what CHC-UFPR is.

In line 8 (39-40 in the updated document), the sentence was rewritten to maintain the number of words requested by the journal. The abbreviation CHC-UFPR has been expanded to its full form in M&M section.

BACKGROUND: Lines 49-50 should be referenced.

In the updated document, the referenced lines 49-50 have been renumbered to 83-84. These lines have been appropriately referenced (7-9).

RESULTS: What are these higher inflammatory test values?

The term "inflammatory tests" encompasses several nonspecific markers employed to evaluate the intensity of the inflammatory process. The markers evaluated were lactate dehydrogenase, ferritin, C-reactive protein, procalcitonin, and D-dimer, which are introduced in line 253-255 of this manuscript. Of note, the specific inflammatory tests under analysis are concisely outlined in Table 3, immediately after the sentence above.

DISCUSSION: Line 292 would better read--patients ages.

Line 292 (301-303 in the updated document) has been revised as suggested for improved clarity.

Reviewer #3: 

I have not concerns concerning the manuscript. I like the fact that datas gotten from the Research project are available. Though I would have appreciated that the Research is extended beyond Brazil. 

Acknowledging the reviewer's point, the pandemic significantly influenced healthcare services in our country, with Brazil experiencing one of the highest mortality rates. In our hospital, we established a distinctive protocol for consistent data collection and laboratory tests, even during the most critical periods. This protocol enabled the execution of the survey detailed in this manuscript. Regrettably, this approach was not universally adopted by other healthcare facilities, posing challenges for extending our findings beyond Brazil.

---

## [Editor Report · Decision Letter 1]

27 Nov 2023

Assessing the impact of the Gamma variant on COVID-19 patient admissions in a Southern Brazilian tertiary hospital – A comparison of dual pandemic phases

PONE-D-23-26555R1

Dear Dr. Raboni,

We’re pleased to inform you that your manuscript has been judged scientifically suitable for publication and will be formally accepted for publication once it meets all outstanding technical requirements.

Kind regards,

Olatunji Matthew Kolawole, Ph.D.

Academic Editor

PLOS ONE
---

## [Editor Report · Acceptance letter]

30 Nov 2023

PONE-D-23-26555R1 

Assessing the impact of the Gamma variant on COVID-19 patient admissions in a Southern Brazilian tertiary hospital – A comparison of dual pandemic phases 

Dear Dr. Raboni:

I'm pleased to inform you that your manuscript has been deemed suitable for publication in PLOS ONE. Congratulations! Your manuscript is now with our production department. 

Kind regards, 

on behalf of

Dr. Olatunji Matthew Kolawole 

Academic Editor

PLOS ONE